# Characterization of a Disease-Suppressive Isolate of *Lysobacter enzymogenes* with Broad Antagonistic Activity against Bacterial, Oomycetal and Fungal Pathogens in Different Crops

**DOI:** 10.3390/plants12030682

**Published:** 2023-02-03

**Authors:** Christian Drenker, Doris El Mazouar, Gerrit Bücker, Sonja Weißhaupt, Eveline Wienke, Eckhard Koch, Stefan Kunz, Annette Reineke, Yvonne Rondot, Ada Linkies

**Affiliations:** 1Julius Kühn-Institute (JKI), Federal Research Centre for Cultivated Plants, Institute for Biological Control, 69221 Dossenheim, Germany; 2Department of Crop Protection, Hochschule Geisenheim University, 65366 Geisenheim, Germany; 3Bio-Protect GmbH, 78467 Konstanz, Germany

**Keywords:** *Lysobacter*, biocontrol, antagonism, BCA, biological fungicide, plant protection

## Abstract

Although synthetic pesticides play a major role in plant protection, their application needs to be reduced because of their negative impact on the environment. This applies also to copper preparations, which are used in organic farming. For this reason, alternatives with less impact on the environment are urgently needed. In this context, we evaluated eight isolates of the genus *Lysobacter* (mainly *Lysobacter enzymogenes*) for their activity against plant pathogens. In vitro, the investigated *Lysobacter* isolates showed broad antagonistic activity against several phytopathogenic fungi, oomycetes and bacteria. Enzyme assays revealed diverse activities for the tested isolates. The most promising *L. enzymogenes* isolate (LEC) was used for further detailed analyses of its efficacy and effective working concentrations. The experiments included in vitro spore and sporangia germination tests and leaf disc assays as well as ad planta growth chamber trials against *Alternaria solani* and *Phytophthora infestans* on tomato plants, *Pseudoperonospora cubensis* on cucumbers and *Venturia inaequalis* on young potted apple trees. When applied on leaves, dilutions of a culture suspension of LEC had a concentration-dependent, protective effect against the tested pathogens. In all pathosystems tested, the effective concentrations were in the range of 2.5–5% and similarly efficacious to common plant protection agents containing copper hydroxide, wettable sulphur or fenhexamid. Thus, the isolate of *L. enzymogenes* identified in this study exhibits a broad activity against common plant pathogens and is therefore a promising candidate for the development of a microbial biocontrol agent.

## 1. Introduction

Plant diseases caused by bacterial and fungal pathogens can lead to severe yield losses in many crops. Annually, around 20% of all harvests are lost due to plant diseases, mainly caused by fungal pathogens [1]. The efficient control of fungal plant diseases is therefore a prerequisite for maintaining high product quality and ensuring food supply. To reach these goals, growers currently mainly depend on the application of chemical synthetic fungicides. Due to their negative impact on the environment and on human and animal health, the use of chemical synthetic fungicides is increasingly questioned [2,3]. Further, the repeated use of these compounds can lead to the development of pathogen populations that are resistant to single or multiple fungicide classes, which leads to a reduction or loss in the effectiveness of these compounds, an issue with increasing importance worldwide [4,5]. In organic agriculture, the application of chemical synthetic fungicides is prohibited; however, most associations of organic growers allow the application of copper products, particular to control plant diseases caused by oomycetes. Despite its approval in organic crop production, copper has proven negative effects on the environment, e.g., it accumulates in soil and has a negative impact on soil life, aquatic biota and biodiversity. Therefore, there is a strong demand to minimize the use of copper, while maintaining high plant protection levels [6].

Microbial biocontrol agents (BCAs) or microbial metabolites with antifungal or antibacterial properties are promising alternatives to chemical synthetic and copper fungicides; their use therefore receives increasing attention [7,8]. Several BCAs have been described and are being applied in agricultural crop production. They either show direct action against plant pathogens or work indirectly by promoting plant growth and fitness. Bacteria and fungi belonging to the first group must be registered as pesticides, while those with indirect activity are generally registered as biofertilizers, plant strengtheners or soil improvers [9]. Some genera, such as *Bacillus*, *Pseudomonas* and *Trichoderma*, have been intensively studied and reviewed regarding their effectiveness and mode of action [10,11,12], while other genera have been less well explored. Among the latter, several taxa are gaining increasing attention, such as the bacterial genus *Lysobacter*, which currently comprises 66 species [13]. *Lysobacter* spp. are gram-negative, show gliding motility and occur ubiquitously in diverse environments, for instance in agricultural soils, water, volcanic ash and plant surfaces such as the rhizosphere [14,15]. The genus name is based on their lytic activity against several microorganisms, such as bacteria, fungi, oomycetes and algae [16]. For the species *L. enzymogenes*, diverse modes of action, for example, the production of toxins and hydrolytic enzymes, have been described in detail and reviewed recently [17]. Several species of the genus *Lysobacter* were reported as interesting candidates for biocontrol purposes to suppress plant diseases [18]. For example, *L. capsici* showed activity against downy mildew on grapevine, *Fusarium oxysporum* on tomato plants and the damping-off of sugar beets [19,20,21,22], *L. antibioticus* controlled *Xanthomonas oryzae* on rice [23], and *L. enzymogenes* effectively suppressed *Fusarium* spp. on cereals and *Pythium aphanidermatum* on cucumbers [24,25]. Taken together, the genus *Lysobacter* has a high potential to be used as a biocontrol agent in diverse crops against different plant pathogens.

To our knowledge, there are no published studies on the effects of *L. enzymogenes* on non-target organisms. However, at least for one isolate of the closely related species *L. capsici*, such tests were conducted according to the relevant Society of Environmental Toxicology and Chemistry (SETAC) guidance document for the registration of biocontrol agents. It was shown that the studied isolate of *L. capsici* had no negative impacts on the following non-target organisms: *Aphidius rhopalosiphi* (parasitic wasp), *Typhlodromus pyri* (predatory mite), *Daphnia magna* (aquatic crustacean) and *Selenastrum carpricornutum* (alga) as well as *Eisenia fetida* and *Enchytraeid albidus* (earthworms). Application on grapevine against *Plasmopara viticola* did not impair the processing (fermentation) of grapes or the quality of the resulting wine [22].

In our study, a set of *Lysobacter* isolates was genotyped and compared regarding their general suppressive activity against several bacterial, oomycetal and fungal pathogens and their enzymatic activity in vitro. The suppressive capacity of a very potent isolate was further characterized in more detail in vitro and ad planta in different crops against different fungal and oomycetal diseases. The aim of our project was to identify highly effective isolates of *Lysobacter* spp. with potential as biocontrol agents in crop protection, in particular for the control of oomycetal diseases.

## 2. Results

### 2.1. Screening of Different Lysobacter sp. Isolates

Of the eight isolates of *Lysobacter* sp. genotyped, six were identified as *L. enzymogenes* based on the 16S rRNA locus. Although exclusive molecular genotyping of *Lysobacter* spp. requires the sequencing of at least two different loci for unambiguous species determination, the results of the 16S rRNA sequencing are listed with the according species names in Table 1, Table 2 and Table 3. This was done because the results of all molecular analyses were similar to the results of a fatty acid methyl ester gas chromatography (MIDI Sherlock MIS) determination carried out previously (data not shown). One isolate (BI-6067) could only be determined at the genus level. In the case of another isolate (Nr. 31, Wolf), the determination remained without a conclusive result, with the highest probability for being *L. antibioticus*, or *L. capsici* or *L. gummosus*, or *L. ginsengisoli*. The enzyme activity of the isolates of *Lysobacter* spp. was analyzed in a qualitative plate assay. All isolates exhibited identical enzyme activities, namely activity of amylase, cellulase, chitinase, glucanase, lipase, protease and xylanase. In contrast, none of the isolates showed siderophore activity or the ability to solubilize phosphate, at least under the tested conditions.

The isolates were then tested regarding their inhibitory action against fungal and oomycetal pathogens (Table 1) as well as bacterial pathogens (Table 2) in vitro. The goal was to select the most promising candidate for further analyses. Generally, the inhibitory action of the tested *Lysobacter* isolates was very broad. The growth of several fungal, oomycetal and bacterial pathogens was inhibited to different extents and in an isolate-specific manner. Pathogenic fungi whose growth was inhibited belonged to different taxonomic groups, such as ascomycetes (*Alternaria radicina*, *Alternaria solani*, *Bipolaris sorokiniana*, *Fusarium culmorum*, *Botrytis cinerea*, *Ascochyta fabae*, *Phoma lingam*), basidiomycetes (*Rhizoctonia solani*) and oomycetes (*Phytophthora infestans*, *Pythium ultimum*). *Alternaria radicina* was only inhibited by isolates BI-6457 and BI-6447. The tested *Lysobacter* isolates suppressed the bacterial phytopathogens *Clavibacter michiganensis*, *Erwinia amylovora* and *Xanthomonas campestris*. The two species of *Pseudomonas* sp. were not inhibited by any of the *Lysobacter* isolates. The inhibitory action against bacterial organisms differed between the tested *Lysobacter* isolates. Isolate BI-6432/2 Kc (*L. enzymogenes*; LEC) was identified as one of the isolates with activity against fungal, oomycetal and bacterial organisms under the tested conditions. Furthermore, this isolate was easy to maintain in solid and liquid cultures on different culture media and able to survive storage in the freezer (data not shown). Therefore, it was used for all further analyses, and is abbreviated as LEC throughout the manuscript.

### 2.2. In Vitro Activity of Lysobacter enzymogenes Isolate LEC

The effects on the sporangia release of *Pseudoperonospora cubensis* and the germination of sporangia of *Phytophthora infestans* were tested in a liquid culture in microtiter plate format (Figure 1). The effects on the spore germination of *Venturia inaequalis* were assessed on glass slides (Figure 2).

LEC was able to reduce the sporangia release of *Ps. cubensis* in all tested concentrations (0.8% to 50%), but none of them were as effective as the copper-containing Cuprozin progress treatment (Figure 1A). LEC led to a complete suppression of the germination of *Ph. infestans* sporangia at concentrations between 50 and 3.1% and was equally as effective as the chemical standard Cuprozin progress. LEC at concentrations as low as 1.6% was still able to significantly reduce the germination rate (Figure 1B).

The germination of *V. inaequalis* spores was completely inhibited at the concentrations of 1.56% and 0.78% (Figure 2). The concentration of 0.39% still led to a significant reduction, while 0.19% did not inhibit germination anymore.

The effect of LEC was tested on strawberry leaves against *Botrytis cinerea* (Figure 3) and on grapevine leaf discs against *Plasmopara viticola* (Figure 4). While its effectiveness against *B. cinerea* was still comparable to the chemical fungicide Teldor (fenhexamid) at the concentration of 6.25%, lower concentrations (3.13% and 1.56%) were not able to suppress disease symptoms. On grapevine leaf discs, an LEC concentration between 5 and 0.5% was equally as effective as the chemical treatment Cuprozin progress (copper hydroxide) against *Pl. viticola*, lower concentrations of 0.1% still had a significant suppressive effect and 0.01% did not have any effect. Taken together, LEC was able to suppress all tested fungal and oomycetal pathogens in different in vitro assays in a concentration-dependent manner.

### 2.3. Ad Planta Activity of Lysobacter enzymogenes Isolate LEC

To investigate the transferability of the in vitro results to intact plants and practical use in horticulture, the disease-suppressive capacity was tested by the application of liquid culture suspensions of LEC on leaves prior to inoculation with oomycetal and fungal pathogens.

The disease-suppressive ability against oomycetal pathogens was tested against *Ps. cubensis* on cucumber plants (Figure 5) and against *Ph. infestans* on tomato plants (Figure 6) in growth chamber assays. On cucumber, concentrations as low as 2.5–5% were able to control the disease symptoms of *Ps. cubensis* comparable to the chemical standard Cuprozin progress. Lower concentrations still led to significant disease suppression. The LEC concentration of 0.5% did not show any disease-suppressive effect and symptoms were comparable to the untreated control. On tomato plants, LEC treatment reduced the disease incidence of *Ph. infestans* comparable to the chemical standard Cuprozin progress, even when applied at concentrations as low as 1%.

The effectiveness of LEC against ascomycetes was investigated for *A. solani* on tomato (Figure 7) and *V. inaequalis* on apple plants (Figure 8). Against *V. inaequalis*, an application of LEC at concentrations as low as 2% resulted in clear inhibitory effects, comparable to treatments with wettable sulphur. The liquid cultures of LEC were also effective against *A. solani* on tomato and able to reduce the disease symptoms in tomato plants at concentrations as low as 1%.

Taken together, LEC was able to suppress the disease incidence ad planta of both oomycetal and fungal plant pathogens comparable to chemical standard treatments and in a concentration-dependent manner. The lowest effective concentrations were between 0.5 and 2%, depending on the investigated pathosystem.

## 3. Discussion

This study investigated the possibilities of using isolates of *Lysobacter enzymogenes* as alternative biocontrol agents to suppress plant pathogens on different crops. The general ability of the genus *Lysobacter* to suppress plant pathogens is well established and has been reviewed recently [17].

The antagonistic potential of bacterial biocontrol agents can be affected by their enzymatic activities [26,27]. Hydrolytic enzymes secreted by antagonistic bacteria, such as cellulase, chitinase, glucanase and protease, that degrade plant pathogenic bacteria or fungi have been described in the context of biocontrol activity [14,27,28]. In our study, a broad spectrum of enzymatic activity (amylase, cellulase, chitinase, glucanase, lipase, protease, xylanase) was observed for the eight tested *Lysobacter* isolates. For *L. enzmogenes*, the species name “enzymogenes” already suggests high enzymatic activity [16,29,30]. The lytic effect of secreted endopeptidases in *Lysobacter* sp. has been shown by Vasilyeva et al. [27] and proposed as a major reason for its ability to compete with both gram-positive and gram-negative bacteria in nature. None of our isolates exerted siderophore or phosphate solubilization activity. Evidence for the phosphate solubilization for *Lysobacter* sp. has not been published so far, while siderophore production has been described for *Lysobacter* sp. [14]. Enzymatic activity was not investigated in more detail here, since the isolates did not differ regarding their activity and therefore this factor did not influence the choice of the most promising isolate. Nevertheless, the enzymatic activity might have contributed as a mode of action to the antagonistic characteristics observed in vitro and ad planta. In addition to the enzymatic activity, the production of antimicrobial metabolites, e.g., dihydromaltophilin (known as heat stable antifungal factor (HSAF)) and derivates thereof, as well as of WAP-8294A, Lysobacteramide A, Lysobactin and other metabolites, has been described as a main mode of action of the genus *Lysobacter* [31,32,33,34,35]. In our experiments, liquid cultures containing both intact cells as well as the metabolites contained in the culture were applied to the plants. Therefore, it cannot be discriminated at this point if the observed suppressive activity was caused by living cells, through metabolic activity or a mix of both.

There was a broad inhibitory activity in vitro by all *Lysobacter* isolates against all tested fungi and oomycetes except for *Alternaria radicina* where the activity was isolate-specific. This is comparable with other *L. enzymogenes* isolates tested in different studies, where an inhibitory effect was observed against the following pathogens: *Aphanomyces cochlioides*, *Bipolaris sorokiniana*, *Fusarium graminearum*, *Fusarium oxysporum*, *Fusarium solani*, *Phytophthora capsici*, *Phytophthora infestans*, *Phytophthora sojae*, *Pythium aphanidermatum*, *Pythium ultimum*, *Rhizoctonia solani* and *Sclerotinia sclerotiorum* [25,36,37,38,39]. Chen et al. [14] described the activity of *L. enzymogenes* against *Alternaria alternata*, but not against other *Alternaria* species. To our knowledge, the observed in vitro activity of *L. enzymogenes* against the *Alternaria* species tested in this study, as well as against *Botrytis cinerea*, has not been described before.

While all our isolates exhibited similar inhibition patterns against the bacterial plant pathogens, different isolate-specific inhibitions were observed in other studies [14]. We observed bactericidal activity against *Clavibacter michiganensis*, *Erwinia amylovora* and *Xanthomonas campestris*, but not against the two species of *Pseudomonas* tested. This is in contrast to Chen et al. [14], who observed an inhibition of *Pseudomonas syringae* pv. *tabaci* by *L. enzymogenes* isolate LE16. Contrarily, Jochum et al. [25] and Li et al. [33] observed no suppressive activity of *L. enzymogenes* either against *Pseudomonas syringae* pv. *tomato* or against *Xanthomonas campestris*, but against *Clavibacter michiganensis.* Differences in antagonistic activity might be isolate-specific, as observed in our studies for fungal diseases (Table 1). It has to be noted that our trials, both on antifungal and on antibacterial activity, were carried out at one temperature and with one culture medium only. Changes in one or in both conditions might have influenced the results.

Although *Lysobacter* has been studied for its efficacy against phytopathogens and has been suggested as a potential candidate for a BCA multiple times, most studies regarding its antagonistic activity have been almost exclusively carried out in vitro. There are only a few studies in the literature on the efficacy ad planta so far, none of which have been conducted with the phytopathogens tested here. In vivo/ad planta analyses are of high importance in the evaluation of biocontrol agents, because efficacy in in vitro tests does not directly correspond to efficacy on plants [40,41]. For *L. enzymogenes*, there are ad planta studies showing efficacy against *Py. aphanidermatum* on cucumber [24,36], *Aphanomyces cochlioides* [37] on sugar beet, *Ph. capsici* on *Capsicum annum* [42], *Fusarium graminearum* on wheat [25] and *Bipolaris sorokiniana* on tall fescue [39]. Ad planta tests against *Pl. viticola* on grapevine with the closely related species *L. capsici* were conducted by [21,22].

Therefore, the ad planta tests with the isolate LEC against *A. solani* and *Ph. infestans* on tomato, *Ps. cubensis* on cucumber and *V. inaequalis* on apple plants reveal new insights regarding the range of antagonism against phytopathogens under scenarios closer to field conditions than in in vitro tests.

*Lysobacter enzymogenes* isolate LEC is a sub-isolate from isolate U407, which was selected in a screening of rhizobacteria for activity against the oomycete *Py. ultimum* on cucumber and kale [43]. The activity of LEC against *Ph. infestans* on tomato and *Pl. viticola* on grapevine leaf discs observed in the present study is in line with this and further corresponds to the results of Puopolo et al. [44], who describe the activity of *Lysobacter capsici* against both oomycetal species. Therefore, the range of suppressive activity in *Lysobacter* spp. against oomycetes seems to be genus-wide.

In most of our in vivo trials, a broad range of LEC concentrations was applied in the plant trials to be able to answer questions related to general effects and to find economically feasible application rates for a commercial product. In our study, LEC concentrations with comparable disease-suppressive action as chemical agents varied (0.5–3% in vitro, 2.5–5% ad planta), which is comparable to the results of Jochum et al. [25]. In vitro, only minor variations in effectiveness between single replicates were observed. In the in vivo trials, however, in some cases differences in effective concentrations were apparent between independent replicates, as shown, e.g., for LEC efficacy against downy mildew on cucumber (Figure 5). Yet, effects were in a similar range and a dose-dependent effect was always apparent. Differences in the experimental conditions such as variable infection pressures cannot be avoided in such a complex test system and do not change the overall conclusion of our experiments.

Whether the observed effective concentrations are economically feasible for commercial use needs to be assessed in further investigations. Preliminary analyses suggest economic concentrations in a range of 1–3% (data not shown). Because the active working concentrations ad planta were in a similar range, the simultaneous control of different pathogens, e.g., *A. solani* and *Ph. infestans* on tomato, may be possible. Moreover, the efficacy of LEC against *Ph. infestans* at 15 °C and *A. solani* at 21 °C suggests applicability over an extended temperature range. In other studies, *Lysobacter* spp. were effective in suppressing plant diseases at temperatures between 18 °C und 25 °C [24,44]. No phytotoxicity was observed at the effective concentrations in all tested crops. Further, to our knowledge, neither the genus *Lysobacter* nor any of its metabolites have been described in the context of human diseases. Both characteristics are a prerequisite for the application in the open field and for the development of a registered plant protection agent.

In the developmental process of a new biocontrol agent, the next mandatory steps would be the up-scaling of production from Erlenmeyer flasks to bigger volumes in fermenters, analysis of shelf-life, and improvement of product processing and formulation, as well as field trials under practical agricultural conditions. The evaluation of the economic feasibility also includes the analysis of parameters such as yield or plant weight and other long-term effects of *L. enzymogenes* that have not been analyzed yet. Moreover, an analysis regarding metabolites produced by the isolate LEC is of interest, both for academic reasons and for the approval as a biocontrol agent.

Taken together, our data confirm the described effectiveness of *L. enzymogenes* against several plant pathogens. With LEC, a suitable isolate of *L. enzymogenes* for biocontrol purposes with a broad spectrum of activity ad planta was identified.

## 4. Materials and Methods

Unless otherwise stated, all chemicals and media ingredients were obtained from Carl Roth (Karlsruhe, Germany).

### 4.1. Cultivation of Bacterial Isolates

#### 4.1.1. *Lysobacter* spp.

For this study the eight *Lysobacter* isolates listed in Table 3 were used. Isolate BI-6432/2 Kc, abbreviated as **LEC** throughout the text, was used for all in vitro and ad planta experiments, which followed screening.

**Table 3 plants-12-00682-t003:** Isolates of *Lysobacter* used in this study, including their origin. For isolate Nr. 31, Wolf, molecular analysis revealed affiliation to be either one of the three species listed in the table with equal probability. JKI-BI: culture collection of the JKI Institute for Biological Control.

Lysobacter Species	Isolate	Source
*Lysobacter* sp.	BI-6067	JKI-BI, cabbage roots, greenhouse
*L. enzymogenes*	BI-6432/1 Kg	JKI-BI, cabbage roots, greenhouse
*L. enzymogenes*	BI-6432/2 Kc (**LEC**)	JKI-BI, cabbage roots, greenhouse
*L. enzymogenes*	BI-6434	JKI-BI, cabbage roots, greenhouse
*L. enzymogenes*	BI-6445	JKI-BI, cabbage roots, greenhouse
*L. enzymogenes*	BI-6447	JKI-BI, cabbage roots, greenhouse
*L. enzymogenes*	BI-6457	Cabbage roots, field, Darmstadt
*Lysobacter* sp.	Nr. 31, Wolf	Georg-August-University of Göttingen, Plant Pathology and Crop Protection Section

The *Lysobacter* isolates were taken from the culture collection of the Institute of Biological Control (JKI-BI), and they were isolated from cabbage roots [43,45,46], except for Isolate Nr. 31 Wolf, which was kindly provided by G. Wolf, Georg-August-University of Göttingen, Plant Pathology and Crop Protection Section. For short-term maintenance, all isolates were cultured alternating on nutrient agar (NA, 8 g L^−1^ Nutrient Broth, 18 g L^−1^ agar, deionized water ad 1 L) and soybean flour glucose calcium chloride agar (SGCA: 8 g L^−1^ soybean flour (Alnatura, Darmstadt, Germany), 7.89 g L^−1^ glucose, 0.95 g L^−1^ CaCl_2_ × 2 H_2_O, 18 g L^−1^ agar, deionized water ad 1 L, adjusted to pH 9 with 2 M NaOH).

*Lysobacter enzymogenes* liquid cultures for the in vitro and in vivo trials were prepared as follows (if not described otherwise): *Lysobacter* was precultured for 4 d at 20 °C on nutrient agar. The inoculum was prepared by suspending a loopful (5 mm) of LEC cells from an NA medium in 5 mL aqueous 0.6% (*w*/*v*) NaCl solution. The OD_660 nm_ was measured and the bacteria concentration was calculated with the formula $bacteria\;mL^{-1}=3.5\times 10^{9}\times OD_{660\;nm}$. This relationship between OD_600_ nm and colony forming units (cfu’s) was investigated experimentally. A total of 50 mL SGCB (8 g L^−1^ soybean flour, 7.89 g L^−1^ glucose, 0.95 g L^−1^ CaCl_2_ × 2 H_2_O, and deionized water ad 1 L, adjusted to pH 9 with 2 M NaOH before autoclaving) as described in Tang et al. [47]), were inoculated with 10^7^ cells per mL and incubated in 250 mL Erlenmeyer flasks for 72 h at 25 °C and 180 rpm in an orbital shaking incubator. In different experimental runs, the resulting liquid cultures had little differences in yield, ranging between 1 and 2.7 × 10^10^ colony forming units per mL, and therefore CFU concentration was not determined for each trial individually.

#### 4.1.2. Phytopathogenic Bacteria

For this study, five phytopathogenic bacteria were used (Table 4). All isolates were obtained from Göttinger Sammlung Phytopathogener Bakterien (GSPB), except for the isolate of *Erwinia amylovora* that was kindly provided by Dr. Esther Moltmann (LTZ Augustenberg, Germany). The bacteria were grown on 0.1 strength tryptic soy agar (TSA) except for *Xanthomonas campestris*, which was cultivated on yeast extract dextrose calcium carbonate agar (YDC, (10 g L^−1^ yeast extract, 20 g L^−1^ glucose, 20 g L^−1^ CaCO_3_, 18 g L^−1^ agar and deionized water ad 1 L).

### 4.2. Cultivation of Fungal and Oomycetal Isolates

For this study, the phytopathogenic fungi and oomycetes listed in Table 5 were used.

All fungi were cultivated on potato dextrose agar (PDA), except for the isolates of *Alternaria* spp. and *Venturia inaequalis*. *Alternaria* spp. were cultivated alternating on vegetable juice (V8) agar [48] and rye B agar [49] and *V. inaequalis* was maintained on leaves of potted apple plants (“Jonagold”) cultivated in a greenhouse [50]. *Phytophthora infestans* was cultivated alternating on V8 agar and Rye B agar. *Pythium ultimum* was cultivated on oatmeal agar (OA, 30 g L^−1^ instant oatmeal (Peter Kölln GmbH & Co. KGaA, Elmshorn, Germany), 15 g L^−1^ agar, deionized water ad 1 L). *Pseudoperonospora cubensis* was maintained as a permanent culture ad planta on cucumber (“Chinese Slangen”) [51] and *Plasmopara viticola* as permanent culture on grapevine plants (“Müller-Thurgau”) [52].

### 4.3. Cultivation of Plants

Cucumber plants (*Cucumis sativus* “Chinese Slangen”, Weigelt Samen, Grolsheim, Germany) were sown four weeks in advance of the plant trials. The substrate consisted of three parts of ProLine Potgrond 30% TerrAktiv (023) substrate (Klasmann-Deilmann GmbH, Geeste, Germany) and one part sand. Per pot (8 × 8 × 8.5 cm), 12 seeds were sown. After one week, single plants were transferred to separate pots of the same size filled with the same medium. Fertilization started 10 days after transplanting with weekly alternating applications of 0.75% (*v*/*v*) AminoVital (Biofa, Münsingen, Germany) and 0.3% (*w*/*v*) Hakaphos rot (Compo Expert, Münster, Germany), respectively.

Tomato plants (*Solanum lycopersicum* “Red Robin”, Weigelt Samen, Grolsheim, Germany) were grown as described for cucumbers except that sowing was five weeks ahead of the trials and with seedling transplanting after two weeks. Plants were fertilized weekly with 0.2% (*w*/*v*) Hakaphos blau (Compo Expert, Münster, Germany).

Cucumber and tomato plants were grown under fluorescent lights (Philips Master TL-D 36W/840, Eindhoven, Netherlands) with a 45 cm distance to the plants (PPFD 156 µmol m^−2^ s^−1^/PAR 35 W m^−2^, light regime 16/8 h, 21 °C).

Grapevine plants (*Vitis vinifera* “Müller-Thurgau”) were cultivated as described in Rondot and Reineke [52]. Briefly, cuttings from mature plants of Geisenheim-University-owned vineyards were soaked in a 0.5% (*v*/*v*) Chinoplant (140 g L^−1^ 8-Hydroxyquinoline sulfate, Cheminova, Stade, Germany) solution for 12 h for disinfection and stored at 4 °C and 95% rel. humidity. Before rooting, cuttings were soaked for 30 min in water followed by removing the lower bud and the dried stem ends. Plants were rooted in a greenhouse in a 50:50 mixture of perlite and standard substrate ED 73 (Patzer Erden, Sinntal-Altengronau, Germany) at 22–24 °C with irrigation twice a week. After 8–10 weeks, rooted cuttings were transferred into 2 L pots with ED73 substrate. Fertilizer was applied weekly (0.1% (*w*/*v*) Flory 3 Mega, Euflor GmbH für Gartenbedarf, München, Germany) starting from the development of the first leaf.

Potted apple trees (*Malus domestica* “Jonagold”) grafted on M 9 rootstocks were held in a greenhouse at 14–30 °C and a 14/10 h light regime. The trees were fertilized with Hakaphos grün (Compo Expert, Münster, Germany).

### 4.4. Screening of Different Lysobacter sp. Isolates

#### 4.4.1. Genotyping of Isolates

Isolates of *Lysobacter* spp. were grown for 3 days as described in Section 4.1.1. Bacteria were scratched from the agar medium and one inoculation loop volume was taken for further analyses. DNA extraction was carried out using the MOBio Power Soil kit according to the manufacturer’s instructions. DNA quality was validated photometrically using the Nanodrop equipment (Peqlab, Darmstadt, Germany). Barcoding was conducted using the 16S-rRNA primers OL1222 (MGAGTTTGATCCTGGCTCAG) and OL1411 (TGCTGCCTCCCGTAGGAGT) [53]. They are covering the V1-V2 regions of the 16S-rDNA gene and amplify an approximately 450 bp fragment of the16S-rDNA. PCR conditions were as follows: 5 min at 95 °C, 40 cycles of 30 s at 95 °C, 30 s at 50 °C and 30 s at 72 °C, followed by a final extension of 5 min at 72 °C. The size of the PCR fragment was checked by gel electrophoresis. After cleaning the PCR products with the ExoSAP-IT PCR Product Cleanup Reagent (Thermo Fisher Scientific, Waltham, MA, USA), they were sent for sequencing to LGC Genomics (Berlin, Germany). For genotyping, sequencing was performed in both directions. Sequence data were subjected to analysis by the Ribosomal Database Project (RDP) Classifier [54] to assign the most probable genus of each isolate as shown in the results section (Table 1). The analysis was performed to confirm data of a fatty acid methyl ester gas chromatography (MIDI Sherlock MIS) determination carried out previously (data not shown).

#### 4.4.2. Analysis of Enzyme Activity

The eight *Lysobacter* isolates were analyzed regarding their enzymatic activities. This screening was performed for amylase, cellulase, chitinase, ß-1,3-glucanase, lipase, protease, xylanase, phosphate solubilization and siderophore production. The assays were based on specific solid substrate-containing growth media in Petri dishes. Cellulose filter discs (Rotilabo, Ø: 6 mm, thickness: 0.75 mm) were soaked in a liquid culture of *Lysobacter*. Four of these discs, each of a different *Lysobacter* isolate, were placed at the distance of 1 cm from the rim of each Petri dish containing one of the enzyme test media. Each treatment was prepared in triplicate. The evaluation of siderophore production was based on an Chromeazurole S (Sigma Aldrich, Taufkirchen, Germany) containing agar as described in Alexander and Zuberer [55] and phosphate solubilization as described in Nautiyal [56]. The medium for the amylase assay consisted of 10 g L^−1^ soy peptone, 4 g L^−1^ beef extract (Thermo Fisher Scientific, Waltham, MA, USA), 5 g L^−1^ NaCl, 10 g L^−1^ starch, 18 g L^−1^ agar and 1 L deionized water [57] and the medium for the protease assay of 5 g L^−1^ soy peptone, 3 g L^−1^ malt extract, 3 g L^−1^ yeast extract, 10 g L^−1^ skim milk powder, 18 g L^−1^ agar and 1 L deionized water [58]. For the remaining enzymes, the medium was prepared according to the following recipe: 2 g L^−1^ NaNO_3_, 1 g L^−1^ K_2_HPO_4_, 0.5 g L^−1^ MgSO_4_, 0.5 g L^−1^ KCl, 0.2 g L^−1^ soy-peptone, 10 g L^−1^ enzyme-specific substrate, 18 g L^−1^ agar and 1 L deionized water. The enzyme-specific substrates were 10 g L^−1^ carboxymethyl cellulose (for cellulase), 10 g L^−1^ colloidal chitin (for chitinase; prepared from powder from shrimp shells, Sigma Aldrich, Taufkirchen, Germany), 10 g L^−1^ ß-1,3-glucan (for glucanase; The Synergy Company, Moab (UT), USA), 10 mL L^−1^ Tween 80 combined with 0.1 g L^−1^ CaCl_2_ (for lipase) or 10 g L^−1^ xylan from beechwood (for xylanase) [59,60,61,62]. The plates were incubated at room temperature for two weeks except for the plates with substrates for phosphatase and chitinase activity, which were incubated for one month. The assessment of enzyme activity was based on the presence (+) or absence (−) of clearing.

#### 4.4.3. Dual Cultures of *Lysobacter* spp. with Fungal and Oomycetal Pathogens

Liquid cultures of the *Lysobacter* isolates to be tested were prepared as described in section “*Lysobacter* spp.”. Dual cultures were prepared for all fungi and oomycetes shown in Table 5. Agar plugs of 10 mm diameter from the fungi and oomycetes to be tested were placed in the center of a 94 mm Petri dish filled with Rye A agar as described in Caten and Jinks [49] with two deviations: Rye kernels were soaked only for 24 h and the extraction was performed for one hour at 68 °C. Four cellulose filter discs with different *Lysobacter* liquid cultures (Section 4.1.1) were placed at the distance of 1 cm from the rim of each Petri dish containing a mycelial plug in the center. Plates with discs soaked in sterile deionized water served as controls. Each treatment was performed in triplicate. Plates were stored at 20 °C in the dark and evaluated when the Petri dishes containing only the fungal and oomycetal controls were fully colonized. The colonization times for the different species were as follows: 2 d for *Py. ultimum*; 5 d for *B. cinerea*, *F. culmorum* and *R. solani*; 8 d for *B. sorokiniana* and *Ph. infestans*; 9 d for *A. solani;* and 28 d for *As. fabae*, *A. radicina* and *P. lingam*. The diameters of the inhibition zones were measured. Since the inhibition zones were elliptic, an average was determined as the geometric mean (G.M.) of the minor (a) and the major (b) diameter: $G.M.=\sqrt{a\times b}$.

#### 4.4.4. Dual Cultures of *Lysobacter* sp. with Bacterial Pathogens

The same *Lysobacter* isolates that were used for the dual cultures with fungi were tested regarding their ability to inhibit phytopathogenic bacteria in dual cultures. This was performed using the pour plate method [63]. Phytopathogenic bacteria (Table 4) were precultured in NB and added to NA shortly before the solidification of the agar (45 °C) at a final concentration of 10^6^ mL^−1^. *Lysobacter* liquid cultures were streaked over the agar containing the pathogenic bacteria with an inoculation loop. Agar plates containing pathogenic bacteria without *Lysobacter* served as controls. Each treatment was performed in triplicate. Plates were assessed after one week incubation in the dark at 20 °C. Inhibitory activity was determined based on the presence (+) or absence (−) of clearing zones around the *Lysobacter* streaks.

### 4.5. In Vitro Activity of Lysobacter enzymogenes Isolate LEC

#### 4.5.1. Zoospore Release of *Pseudoperonospora cubensis*

The effect of isolate BI-6432/2 Kc (LEC) of *Lysobacter enzymogenes* liquid culture on the zoospore release from the sporangia of *Pseudoperonospora cubensis* was evaluated in vitro in 24-well plates. In each well, 250 µL of a suspension of 2 × 10^5^
*Ps. cubensis* sporangia mL^−1^ in deionized water was mixed with a 250 µL LEC liquid culture twofold concentrated to yield a onefold concentration in the final suspension to be tested. The LEC liquid culture was tested in serial twofold dilutions in final concentrations (*v*/*v*) between 50% and 0.8%. Sporangia suspensions mixed with deionized water, 0.52% (*v*/*v*) Cuprozin progress (standard field application concentration; concentration of active ingredient in product: 383.8 g L^−1^ copper hydroxide; Biofa, Münsingen, Germany and a sterile SGCB medium in final concentrations (*v*/*v*) of 50% and 12.5% served as controls. All treatments were performed in triplicates. Immediately after the complete preparation of all treatments, 50 sporangia per well were assessed microscopically on the release of zoospores (e.g., full/empty sporangia). Plates were incubated at 13 °C to promote sporangia release, and after 24 h a second assessment was performed. From the difference of the two assessments, the relative rate of release was determined.

#### 4.5.2. Sporangia Germination Test with *Phytophthora infestans*

Sporangia of *Phytophthora infestans* were harvested from three-week-old cultures grown on Rye B agar. Therefore, 5 mL of sterile deionized water was added to the cultures, followed by gentle rubbing with a Drigalski spatula. Sporangia suspensions from multiple plates were combined and adjusted to an initial concentration of 5 × 10^5^ sporangia mL^−1^. The effects on *Ph. infestans* sporangia were examined in the same way as described for *Ps. cubensis* (Section 4.5.1), with the difference that the germination rate was assessed instead of the release rate.

#### 4.5.3. Spore Germination Test of *Venturia inaequalis*

Infected leaves with sporulating *Venturia inaequalis* lesions stored frozen at −20 °C were thawed and shaken in tap water to obtain conidial suspensions, which were adjusted to 2 × 10^5^ spores per mL and mixed with the individual test preparations in twofold concentration in equal quantities. The prepared mix (100 μL) was added to the wells of microscope slides and incubated in a humid chamber for 24 h at 20 °C. At least 100 conidia per replicate were analyzed for germination under the microscope and the percentage of germinated conidia was calculated in 10 replicates.

#### 4.5.4. *Botrytis cinerea* on Strawberry Leaves

*Botrytis cinerea* (isolate 222) was grown on ME+ agar (3% malt extract, 0.5% peptone, 0.02% penicillin G, 0.02% streptomycin, all *w*/*v*) at 20 °C for 10–14 days under a fluorescent tube (12 h light, 12 h dark) until the *B. cinerea* culture sporulated. Conidia were washed from the plates with autoclaved deionized water and a concentration of 2 × 10^6^ mL^−1^ was adjusted. Each treatment consisted of 10 detached strawberry leaves. Conidia and dilution series of LEC liquid culture or 0.2% (*w*/*v*) Teldor (500 g kg^−1^ fenhexamid, Bayer CropScience, Monheim, Germany) as chemical standard were mixed in equal quantities (final concentrations: 10^6^ conidia mL^−1^, 0.1% Teldor (practical field application concentrations) or LEC in concentrations of 50, 25, 12.5, 6.25, 3.13 or 1.56%) and incubated for 20 min at 20 °C. Strawberry leaves were injured with a scalpel and a piece of SA agar (0.5% sucrose, 1.5% agar, *w*/*v*) was placed on the incision. A total of 10 µL of the mix of conidia and test preparation was applied by pipetting under the agar plug. Leaves were incubated at 20 °C in a humid chamber, and lesion diameters were measured after 9 days of incubation.

#### 4.5.5. *Plasmopara viticola* on Grapevine Leaf Discs

Leaves of rooted shoots (approx. 3 months) of grapevine (*Vitis vinifera* “Müller-Thurgau”) were harvested and leaf discs (18 mm diameter) were cut with a cork borer. Twelve leaf discs were equally distributed with the abaxial side facing upwards on Petri dishes with water agar (1%, *w*/*v*). Per treatment, 3 Petri dishes were analyzed, 36 leaf discs in total. The leaf discs were sprayed with 0.32% (*v*/*v*) Cuprozin progress (application rate of reference product as under practical field application conditions), sterile SGCB medium or with the indicated concentrations of *L. enzymogenes* (LEC), which was grown in liquid culture as described in Section 4.1.1. The treated leaf discs were left to dry under the sterile bench for one hour. The next day, the leaf discs were spray-inoculated with 10^5^
*Plasmopara viticola* sporangia mL^−1^ until a complete wetness of the leaf disc surface and left overnight at 26 °C. The following day, the plates were opened again and left for drying for one hour under the sterile bench. Six days after inoculation, the leaf discs were sprayed with sterile water to induce sporulation, and the next day disease severity (visual observation of infected leaf area [%]) on each leaf disc was determined. The observed infestation level was classified into 5% increments in the range of 0–10% and 10% increments in the range of 10–100%.

### 4.6. Ad Planta Activity of Lysobacter enzymogenes Isolate LEC

The effects of the *L. enzymogenes* liquid culture on plant pathogens was also assessed ad planta for downy mildew (*Pseudoperonospora cubensis*) on cucumber plants, scab (*Venturia inaequalis*) on apple plants, and early blight (*Alternaria solani*) and late blight (*Phytophthora infestans*) on tomato plants. Plant trials were conducted with the *L. enzymogenes* isolate BI-6432/2 Kc (LEC). LEC was applied as dilutions (expressed in percent) of the liquid culture grown for 72 h. CFU determination was not conducted in every trial, since the CFU ranged between 1 and 2.7 × 10^10^ per ml in all previous trials.

#### 4.6.1. *Pseudoperonospora cubensis* on Cucumber Plants

To get broader insight on the concentration-dependent effect, the LEC liquid culture effectiveness was evaluated against *Ps. cubensis* on cucumber in plant trials with two different ranges of concentrations. Cucumber plants with fully developed third leaves, approximately four weeks old, were sprayed on the abaxial sides of the second and third leaves with the LEC liquid culture until runoff in the concentrations (*v*/*v*) of 10%, 5% and 2.5% in the first trial and 10%, 5%, 3%, 1% and 0.5% in the second trial. As a reference product, 0.52% (*v*/*v*) Cuprozin progress (CU) was used. The application rate represents practical field application conditions. The pathogen-free (CON–) and the pathogen control (CON+) were treated with deionized water. A sterile SGCB medium at a concentration of 10% (*v*/*v*) (SGCB) was also applied as a control to evaluate the intrinsic effect of the culture medium. One day after the application of the test compounds, the plants were sprayed on the abaxial leaf sides with 10^4^
*Ps. cubensis* sporangia ml^−1^ in deionized water, except for the pathogen-free control, which was sprayed with deionized water only. The plants were placed in plant trays covered with hoods to assure high humidity for 24 h in darkness. The subsequent cultivation was without hoods and as described in Section 4.3. After one week, the disease severity was visually rated based on the percentage of infected leaf area and classified as follows: for areas <10%—0%, 1%, 3%, 7% and for areas of 10–100%—5% increments. Per treatment, six plants were used in the first trial and 12 in the second, distributed in two trays. The tray placements were randomized. Two independent trials were conducted.

#### 4.6.2. *Phytophthora infestans* on Tomato Plants

The effect of the LEC liquid culture against *Ph. infestans* on tomato plants was evaluated in plant trials analogously to those with *Ps. cubensis* on cucumbers with the following modifications. The LEC culture was applied in concentrations (*v*/*v*) of 10%, 5% and 1%. Tomato plants had fully developed fifth leaves and were approximately five weeks old. The third to fifth leaves were treated. The sporangia concentration for inoculation was 2 × 10^4^
*Ph. infestans* sporangia ml^−1^ in deionized water. The *Ph. infestans* sporangia were harvested from three-week-old cultures on Rye B agar by flooding with 5 mL deionized water and gentle scraping with a Drigalski spatula two times. Sporangia suspensions from several plates were combined and filtered through one layer of medical gauze. The plants were incubated in darkness for 24 h and covered with transparent hoods for the whole duration of the trials. For optimal development of *Ph. infestans* symptoms, plants were grown at 15 °C, and all other parameters were as described in Section 4.6.1. An assessment of infested leaf area with *Ph. infestans* typical symptoms was conducted after one week (in % increments as described in Section 4.6.1). Per treatment, nine plants were used, divided in three trays, which were placed in a randomized design. The trial was conducted three times.

#### 4.6.3. *Alternaria solani* on Tomato Plants

The effect of the LEC liquid culture against *Alternaria solani* on tomato plants was evaluated as described for *Ph. infestans* on tomato plants (Section 4.6.2). Deviations were as follows: the *A. solani* (A.s.752_3) inoculum had a concentration of 5 × 10^3^ conidia mL^−1^ in deionized water. The mass conidia production of *A. solani* was conducted after the method of Shahin and Shepard [64]. A preculture was prepared by spreading a 100 µL *A. solani* mycelium suspension on V8 Agar. After 4 days, one third of each plate was placed on a sporulation medium (20 g sucrose, 30 g CaCO_3_, 20 g agar, 1 L deionized water, pH adjusted to 7,4 with 1 M HCl). The transferred agar was cut into small pieces and spread over the sporulation medium followed by the addition of 2 mL of sterile deionized water. The plates were incubated for 12 days at 21 °C under alternating cycles (12 h/12 h) of darkness and NUV-light (Philips TL-D 36 W BLB, Eindhoven, Netherlands, 40 cm distance between light source and Petri dishes). Spores were harvested by flooding the plates with 10 mL of 0.125% (*v*/*v*) Tween 80 solution and rubbing with a Drigalski spatula. The resulting spore suspensions were combined and filtered through one layer of medical gauze. The spore concentrations were adjusted to the desired spore concentrations with sterile deionized water. The tomato plants were treated on the abaxial and adaxial sides of the third to fifth leaves. LEC concentrations (*v*/*v*) were 10%, 5%, 3% and 1%. The temperature in the plant growth room was 21 °C. After two weeks, the plants were assessed regarding the leaf area infested with *Alternaria* typical symptoms (in % increments as described in Section 4.6.1). Per treatment, nine plants (eight in the second trial) were used, divided in three trays, which were placed randomized. The trial was conducted two times.

#### 4.6.4. *Venturia inaequalis* on Apple Plants

The experiments were performed as described in Kunz et al. [50], with some modifications. Apple plants were treated with 0.25% (*w*/*v*) wettable sulphur (standard concentration in apple production, Agrostulln GmbH, Stulln, Germany) or a liquid culture of *L. enzymogenes* (LEC) 1 h before pathogen inoculation in the indicated concentrations until runoff. Per treatment, five shoots were sprayed. For inoculation, conidia of V. inaequalis were obtained from defrosted diseased leaves as described in Section 4.5.3. The youngest three unfolded leaves on a shoot were inoculated with 10^5^ conidia mL^−1^ until runoff and further incubated at 18–24 °C and 100% humidity for 20 h. The plants were then kept in the greenhouse. At 20 days after inoculation, the disease incidence severity for each shoot was calculated as the average of the proportion of the diseased leaf area of the three youngest inoculated leaves [50]. Up to ten single spots of sporulating infection sites were counted per leaf and rated as 1% leaf area per spot. If more than 10 spots occurred per leaf, the portion of the symptomatic leaf area was estimated in steps of 10%. The average of the diseased leaf area of 5 shoots per treatment was calculated. The efficiency of the test preparation was calculated for each experiment by comparing the disease incidence severity with the untreated control according to Abbott [65]. The experiment was conducted two times.

### 4.7. Statistical Analysis

The effects of different treatments (e.g., concentrations of LEC or controls) regarding sporangia release/germination or disease severity (e.g., area of infestation on plant leaves) were statistically evaluated and visualized using R (v4.0.3, [66]) in R Studio (v1.4.1103). Due to heteroscedasticity and the lack of normal distribution, data were statistically analyzed with generalized least square regression models (GLS) of the package “nlme” (v3.1-149, [67]). Details on and theory of GLS are described in [68,69,70,71]. The implementation of GLS in R is described in Pinheiro et al. [67]. Treatments where all observed values were zero (e.g., pathogen-free/healthy treatments) had to be transformed by random values between 10^−11^ and 10^−12^ because otherwise GLS regression would not be applicable [72]. Post hoc tests were performed based on Tukey’s test with adjustments for the number of estimates and significance level alpha = 0.05 using the package “emmeans” (v1.5.4, [73]). Boxplots were generated with the package “ggplot2” (v3.3.5, [74]).

## Figures and Tables

**Figure 1 plants-12-00682-f001:**
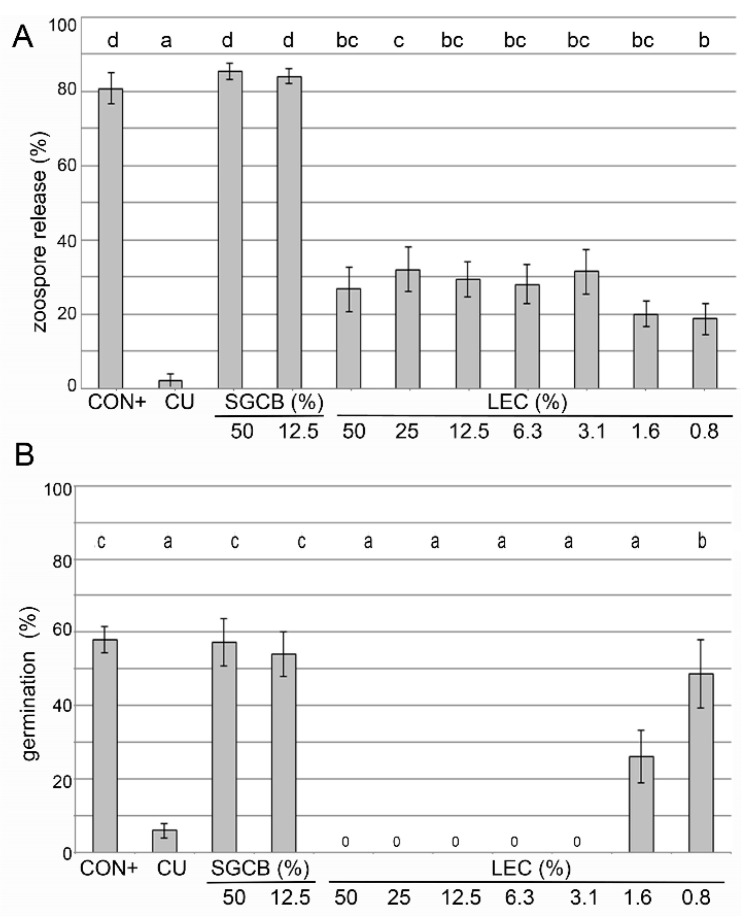
Effect of *Lysobacter enzymogenes* isolate LEC on sporangia of oomycetes. Shown is (**A**): zoospore release of *Pseudoperonospora cubensis* and (**B**): sporangia germination of *Phytophthora infestans*, both after 24 h incubation. Trials were carried out in microtiter plates. CON+: water with pathogen, CU: Cuprozin progress (0.52%), SGCB: sterile soy medium, LEC: *L. enzymogenes* (isolate LEC) liquid culture at the indicated concentrations; *n* = 3, *p*-value < 0.05, data analysis: GLS regression. Different letters indicate significant differences.

**Figure 2 plants-12-00682-f002:**
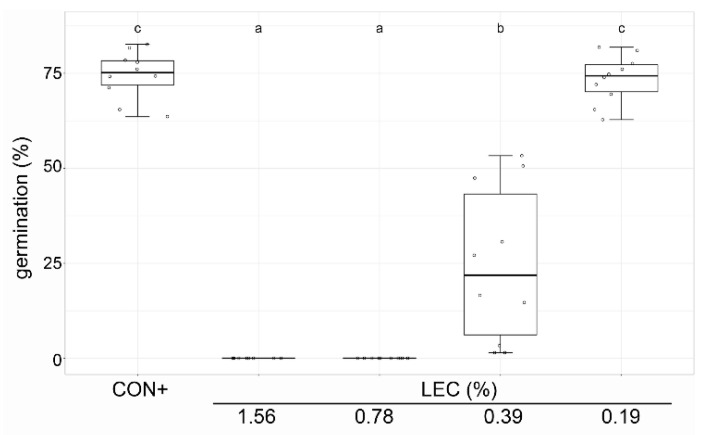
Effect of *Lysobacter enzymogenes* isolate LEC on conidia germination of *Venturia inaequalis*. Shown is conidia germination after 24 h of incubation in *L. enzymogenes* liquid culture at the indicated concentrations at 20 °C on microscope slides. CON+: water treatment with pathogen, LEC: *L. enzymogenes* (isolate LEC) liquid culture in the indicated concentrations; *n* = 10, *p*-value < 0.05, data analysis: GLS regression. Different letters indicate significant differences.

**Figure 3 plants-12-00682-f003:**
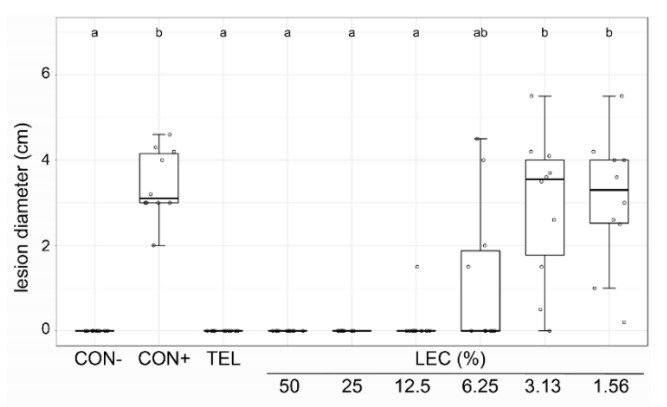
Effect of *Lysobacter enzymogenes* isolate LEC against *Botrytis cinerea* on strawberry leaves. Shown is the lesion diameter 9 days after inoculation of detached strawberry leaves with *B. cinerea*, incubated in a moist chamber at room temperature. CON−: healthy control, CON+: pathogen control, TEL: chemical standard Teldor (0.1%), LEC: *L. enzymogenes* (isolate LEC) liquid culture at the indicated concentrations; *n* = 10 leaves, *p*-value < 0.05, data analysis: GLS regression. Different letters indicate significant differences.

**Figure 4 plants-12-00682-f004:**
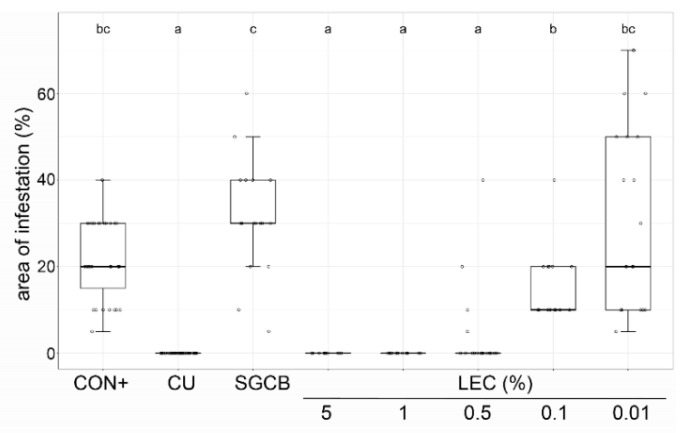
Effect of *Lysobacter enzymogenes* isolate LEC against *Plasmopara viticola* on grapevine leaf discs. Shown is the area of infestation 7 days post inoculation with *P. viticola*. Preventive treatments were conducted 24 h before inoculation, leaf discs were incubated in agar plates at room temperature. CON+: pathogen control, CU: Cuprozin progress (0.32%), SGCB: sterile soy medium, LEC: *L. enzymogenes* (LEC) liquid culture at the indicated concentrations; *n* = 36 leaf discs, *p*-value < 0.05, data analysis: GLS regression. Different letters indicate significant differences.

**Figure 5 plants-12-00682-f005:**
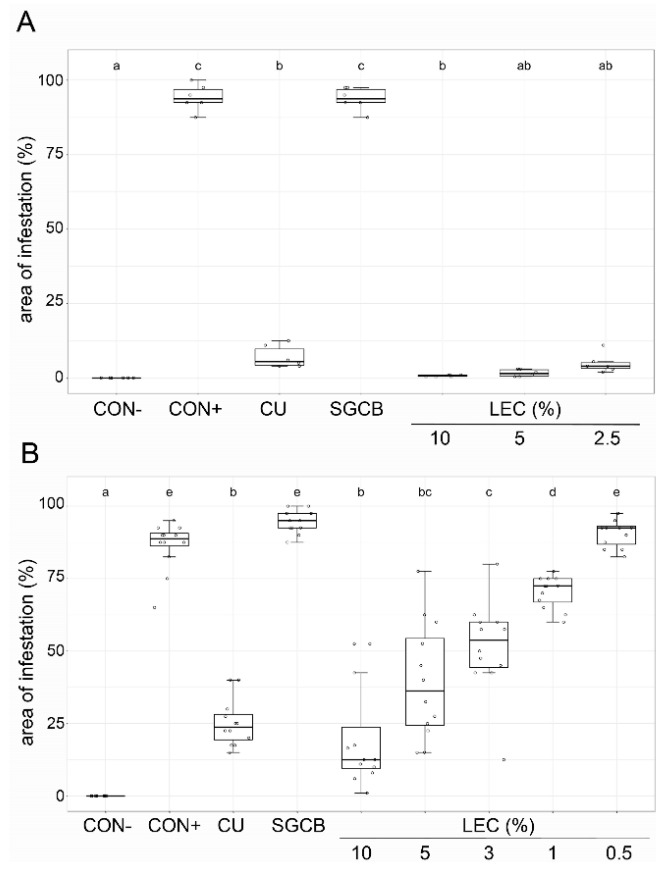
Effect of *Lysobacter enzymogenes* isolate LEC against downy mildew (*Pseudoperonospora cubensis*) on cucumber plants. Shown is the area of infestation one week after inoculation with *P. cubensis*. Treatments were applied 24 h before inoculation. Plants were incubated in the growth chamber at 21 °C. CON−: healthy control, CON+: pathogen control, CU: Cuprozin progress (0.52%), SGCB: sterile soy medium, LEC: *L. enzymogenes* (isolate LEC) liquid culture at the indicated concentrations. Shown are two individual trials with (**A**): LEC concentrations between 10 and 2.5% and (**B**): LEC concentrations between 10 and 0.5%. *n* = 6 (A)/12 (B), *p*-value < 0.05, data analysis: GLS regression. Different letters indicate significant differences.

**Figure 6 plants-12-00682-f006:**
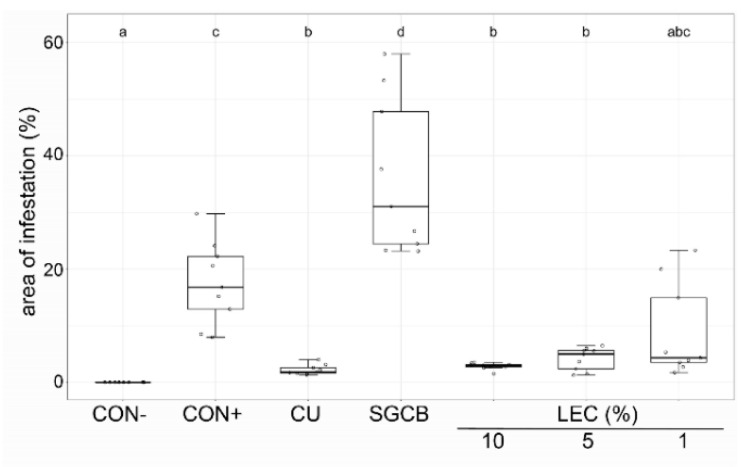
Effect of *Lysobacter enzymogenes* isolate LEC against *Phytophthora infestans* on tomato plants. Shown is the area of infestation one week after the inoculation with *P. infestans*. Treatments were applied preventively 24 h before inoculation. Plants were incubated in the growth chamber at 15 °C. CON−: healthy control, CON+: pathogen control, CU: Cuprozin progress (0.52%), SGCB: sterile soy medium, LEC: *L. enzymogenes* (isolate LEC) liquid culture at the indicated concentrations; *n* = 9, *p*-value < 0.05, data analysis: GLS regression. Different letters indicate significant differences.

**Figure 7 plants-12-00682-f007:**
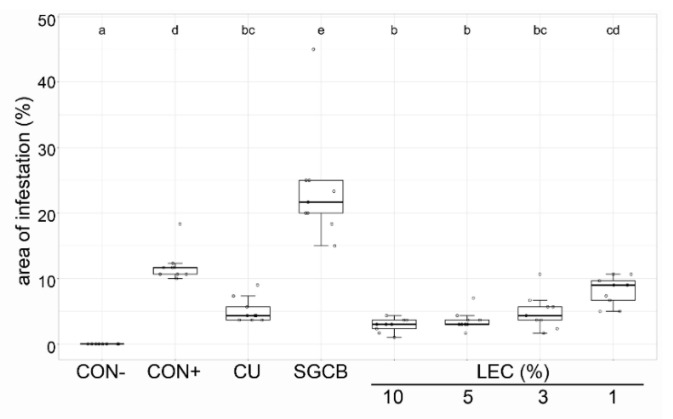
Effect of *Lysobacter enzymogenes* isolate LEC against *Alternaria solani* on tomato plants. Shown is the area of infestation two weeks after inoculation with *A. solani*. Treatments were applied 24 h before inoculation. Plants were grown in the growth chamber at 21 degrees. CON−: healthy control, CON+: pathogen control, CU: Cuprozin progress (0.52%), SGCB: sterile soy medium, LEC: *L. enzymogenes* (isolate LEC) liquid culture at the indicated concentrations; *n* = 9, *p*-value < 0.05, data analysis: GLS regression. Different letters indicate significant differences.

**Figure 8 plants-12-00682-f008:**
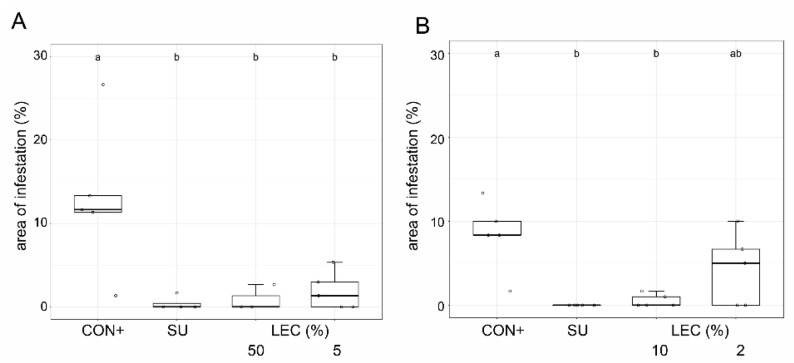
Effect of *Lysobacter enzymogenes* isolate LEC on disease severity (area of infestation) against *Venturia inaequalis* on apple plants (“Jonagold”). Shown is the area of infestation 20 days after treatment and inoculation with scab conidia. CON+: inoculated but untreated control, SU: wettable sulphur (0.25%), LEC: *L. enzymogenes* (LEC) liquid culture at the indicated concentrations. Two individual trials with (**A**): 5 and 50% LEC; (**B**): 10 and 2% LEC; *n* = 5, *p*-value < 0.05, data analysis: GLS regression. Different letters indicate significant differences.

**Table 1 plants-12-00682-t001:** Inhibition of growth of phytopathogenic fungi and oomycetes in dual cultures with isolates of *Lysobacter* spp. *n* = 3.

		Fungal/Oomycetal Pathogen									
		*Ascochyta fabae*	*Alternaria radicina*	*Alternaria solani*	*Botrytis cinerea*	*Bipolaris sorokiniana*	*Fusarium culmorum*	*Phoma lingam*	*Rhizoctonia solani*	*Phytophthora infestans*	*Pythium ultimum*
*Lysobacter* Species	Isolate	Diameter of Inhibition Zone (mm)									
*Lysobacter* sp.	BI-6067	28 ± 3	0 ± 0	21 ± 1	17 ± 2	16 ± 2	24 ± 0	27 ± 1	20 ± 0	31 ± 5	10 ± 0
*L. enzymogenes*	BI-6432/1 Kg	11 ± 0	0 ± 0	13 ± 2	17 ± 5	11 ± 1	10 ± 0	20 ± 4	15 ± 0	36 ± 3	10 ± 1
*L. enzymogenes* (LEC)	BI-6432/2 Kc	43 ± 1	0 ± 0	31 ± 7	42 ± 1	16 ± 1	32 ± 2	40 ± 3	37 ± 2	47 ± 5	12 ± 1
*L. enzymogenes*	BI-6434	39 ± 5	0 ± 0	30 ± 10	45 ± 2	32 ± 2	30 ± 7	31 ± 1	22 ± 3	37 ± 1	10 ± 2
*L. enzymogenes*	BI-6445	29 ± 9	0 ± 0	0 ± 0	28 ± 2	0 ± 0	6 ± 8	0 ± 0	11 ± 15	29 ± 2	0 ± 0
*L. enzymogenes*	BI-6447	35 ± 6	9 ± 13	40 ± 7	37 ± 6	14 ± 0	47 ± 7	43 ± 2	20 ± 4	34 ± 2	10 ± 0
*L. enzymogenes*	BI-6457	42 ± 4	14 ± 2	35 ± 6	37 ± 8	22 ± 6	34 ± 1	47 ± 2	21 ± 1	46 ± 4	12 ± 0
*Lysobacter* sp. **^1^**	Nr. 31, Wolf	32 ± 0	0 ± 0	0 ± 0	36 ± 3	20 ± 4	24 ± 2	30 ± 7	0 ± 0	30 ± 1	0 ± 0

**^1^**: similar probability for L. gummosus, L. ginsengisoli, L. antibioticus and L. capsici.

**Table 2 plants-12-00682-t002:** Inhibition of plant pathogenic bacteria in dual cultures with isolates of *Lysobacter* spp. Inhibition was assessed based on the presence (+) or absence (−) of clear halos around *Lysobcater* colonies. *n* = 3.

		Bacterial Pathogen				
		*Clavibacter* *michiganensis*	*Erwinia amylovora*	*Pseudomonas* *syringae*	*Pseudomonas tabaci*	*Xanthomonas campestris*
Lysobacter Species	Isolate	Inhibition (+: Yes; −: No)				
*Lysobacter* sp.	BI-6067	+	+	−	−	+
*L. enzymogenes*	BI-6432/1 Kg	+	+	−	−	+
*L. enzymogenes* (LEC)	BI-6432/2 Kc	+	+	−	−	+
*L. enzymogenes*	BI-6434	+	+	−	−	+
*L. enzymogenes*	BI-6445	+	+	−	−	+
*L.enzymogenes*	BI-6447	+	+	−	−	+
*L.enzymogenes*	BI-6457	+	+	−	−	+
*Lysobacter* sp.	Nr. 31, Wolf	+	−	−	−	+

**Table 4 plants-12-00682-t004:** Phytopathogenic bacteria used in this study, including their origin. GSPB: Göttinger Sammlung Phytopathogener Bakterien (The Göttingen Collection of Phytopathogenic Bacteria, University of Göttingen, Germany). LTZ: Landwirtschaftliches Technologiezentrum Augustenberg (Dr. Esther Moltmann), Germany.

Species	Isolate	Source
*Clavibacter michiganensis* ssp. *sepedonicus*	GSPB2825	GSPB
*Erwinia amylovora*	E.a.639	LTZ
*Pseudomonas syringae* pv. *phaseolicola*	GSPB1715	GSPB
*Pseudomonas tabaci*	GSPB117	GSPB
*Xanthomonas campestris*	GSPB1386	GSPB

**Table 5 plants-12-00682-t005:** Phytopathogenic fungi and oomycetes used in this study, including their origin. JKI-BI: culture collection of the JKI-Institute for Biological Control.

**Fungal Species**	**Isolate**	**Source**
*Alternaria radicina*	A.rad.1	JKI-BI, carrot Seeds
*Alternaria solani*	A.s.714_1	TU München, Dr. Hausladen, potato “Amado“, Hamersdorf
*Alternaria solani*	A.s.752_3	TU München, Dr. Hausladen, potato “Kuras“, 29,468 Bergen
*Ascochyta fabae*	AF-4Re	JKI-BI/Prof. G. Kahl, University Frankfurt, *Vicia faba*, Iran
*Botrytis cinerea*	B.c.1	JKI-BI, unknown
*Botrytis cinerea*	222	Bio- Protect GmbH, strawberry fruit, Konstanz
*Bipolaris sorokiniana*	BI-7191	JKI-BI, Barley, Ukraine
*Fusarium culmorum*	VIII 18	Kiel University, wheat
*Phoma lingam*	T12aD34	University of Göttingen, oilseed rape
*Rhizoctonia solani*	AG2-2IIIb	Kiel University, maize
*Venturia inaequalis*		Bio- Protect GmbH, apple leaves, Konstanz
**Oomycetal Species**	**Isolate**	**Source**
*Pythium ultimum*	P.u.1	JKI-BI, cress
*Phytophthora infestans*	Syngenta K5509	Syngenta, potato
*Pseudoperonospora cubensis*		JKI-BI, cucumber leaves
*Plasmopara viticola*	isolate mixture	Geisenheim University, mix of five isolates from different origins (Geisenheim, Saulheim, Osann-Monzel, Wackernheim (Germany) and Remich (Luxemburg))

## Data Availability

The data presented in this study are available on request from the corresponding authors.

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
