# Peer review of "Characterization of a Disease-Suppressive Isolate of Lysobacter enzymogenes with Broad Antagonistic Activity against Bacterial, Oomycetal and Fungal Pathogens in Different Crops"

_plants, 2023, doi:10.3390/plants12030682_

Round 1

Reviewer 1 Report

This manuscript is superb from all aspects.  It is difficult to test such a wide array of plant pathogens, especially to be able to successfully induce disease on plants with such different fungi and oomycetes.  This is a very good manuscript.  The authors make a strong case for additional efforts on this group of biocontrol bacteria, and particularly for the one isolate used in extensive plant inoculations.  The only corrections I would recommend is on line 541, solani was misspelled and with the references, some of them are not formatted correctly (28, 29, 30, 31, 49, 60, 62, 63, 66, and 70).

Reviewer 2 Report

The authors study the in vitro antagonistic potential of some Lysobacter isolates. One of the isolates was chosen for a comprehensive investigation including the analysis of the antagonistic activity ad planta. I found merit in this study, however, there are some shortcomings in Material and Methods and the Results which should be revised.

-          The authors stated that they genotyped/identified the Lysobacter isolates. It seems that the taxonomic assignment is based on the 16S rRNA gene analysis. Please be more precise in the results and specify it accordingly or provide further data about the phylogenetic/taxonomic assignment.

-          “Since all molecular analyses confirmed the results of” FAME… Sounds really cryptic, please specify.

-          Tab. 1. It’s unusual to present three species. You can indicate the most probable species name in quotation marks (and an explanation in the table footnote) or designate it as Lysobacter sp.

-          l. 487. Please provide the reference for the used primers as well as the amplified region/amplicon length in the 16S rRNA gene.

-          L. 493. For a taxonomic assignment, a BLAST search is not really suitable. Please use curated databases with the type strains of the bacterial species, eg. RDP classifier and/or EzBioCloud.

-          For the most bacterial genera, an unambiguous assignment of the species only based on the 16S RNA is not possible even if the total gene is sequenced. Please check it carefully using the recommended databases! Again, if the assignment is unsure I recommend to use the most probable species name with quotation marks.

-          Identification of the Lysobacter strains is not one of main subjects of the study. I suggest to omit it in the title.

-          l. 107. You write about a quantitative plate assay but provide qualitative data. Please revise.

-          L. 121-122. Please refer to the taxonomic level (comparable to the fungi), that means please state the phylum instead of the type of the cell wall.

-          Table 1. Statistics are missing! Is the growth inhibition significant against the control? I suggest to use a bacterial strain (of another species) without antagonistic activity for this purpose.

-          L. 537. I think water is not a useful control. In a cocultivation test, there is also a competition for nutrients without any antagonistic activity. Therefore, a bacterial isolate with similar growth rates but without antagonistic activity is much more suitable. This would also be a valid control for the statistical analysis.

Your results are impressive, but you show the short-term effect of the inoculation strain (1-3 weeks). I guess that a practical application is dependent on a longer-term effect on plants? This should be addressed in the discussion.

Round 2

Reviewer 2 Report

The manuscript was substantially improved. I agree that the screening does not need a statistical evaluation.

I do not have any further comments.